# Toxic eburicol accumulation drives the antifungal activity of azoles against *Aspergillus fumigatus*

Hesham Elsaman[1,6], Eveny Golubtsov [1,6], Sean Brazil[2], Natanya Ng [2], Isabel Klugherz [3,4], Ronny Martin [1], Karl Dichtl[3,4], Christoph Müller [5] & Johannes Wagener [1,2,3]

Azole antifungals inhibit the sterol C14-demethylase (CYP51/Erg11) of the ergosterol biosynthesis pathway. Here we show that the azole-induced synthesis of fungicidal cell wall carbohydrate patches in the pathogenic mold *Aspergillus fumigatus* strictly correlates with the accumulation of the CYP51 substrate eburicol. A lack of other essential ergosterol biosynthesis enzymes, such as sterol C24-methyltransferase (Erg6A), squalene synthase (Erg9) or squalene epoxidase (Erg1) does not trigger comparable cell wall alterations. Partial repression of Erg6A, which converts lanosterol into eburicol, increases azole resistance. The sterol C5-desaturase (ERG3)-dependent conversion of eburicol into 14-methylergosta-8,24(28)-dien-3β,6α-diol, the "toxic diol" responsible for the fungistatic activity against yeasts, is not required for the fungicidal effects in *A. fumigatus*. While ERG3-lacking yeasts are azole resistant, ERG3-lacking *A. fumigatus* becomes more susceptible. Mutants lacking mitochondrial complex III functionality, which are much less effectively killed, but strongly inhibited in growth by azoles, convert eburicol more efficiently into the supposedly "toxic diol". We propose that the mode of action of azoles against *A. fumigatus* relies on accumulation of eburicol which exerts fungicidal effects by triggering cell wall carbohydrate patch formation.

Azole antifungals are widely used as biocides, plant protection products, and antimicrobials for chemoprevention, chemoprophylaxis, and treatment of fungal infections in animals and humans[1]. Azoles inhibit the sterol C14-demethylase (CYP51, also known as Erg11 in yeasts) which is one of the key enzymes in the ergosterol biosynthesis pathway[2,3]. In most fungi, ergosterol is the most abundant sterol in the membranes, where it serves a function similar to cholesterol in human and animal cells[4]. Inhibition of CYP51 proved to be highly effective against many fungal species, including human and animal pathogenic

fungi such as dermatophytes as well as the major causative agents of invasive fungal infections such as yeasts in the genera *Cryptococcus* and *Candida* and molds in the genus *Aspergillus*[5]. However, despite their clinical efficacy, the effects of azoles on fungal viability and growth vary greatly among fungal species. Against certain species, especially those in the medically important genus *Candida*, they exert only a growth-inhibitory (fungistatic) activity without ultimately killing the fungal pathogen[6]. Other pathogens, such as the medically important molds in the genus *Aspergillus* are readily killed by azoles which is

[1]Institut für Hygiene und Mikrobiologie, Julius-Maximilians-Universität Würzburg, Würzburg, Germany. [2]Department of Clinical Microbiology, School of Medicine, Trinity College Dublin, the University of Dublin, St James's Hospital Campus, Dublin, Ireland. [3]Max von Pettenkofer-Institute for Hygiene and Medical Microbiology, Faculty of Medicine, Ludwig-Maximilians-Universität München, Munich, Germany. [4]Diagnostic and Research Institute of Hygiene, Microbiology and Environmental Medicine, Medical University of Graz, Graz, Austria. [5]Department of Pharmacy - Center for Drug Research, Ludwig-Maximilians-Universität München, Munich, Germany. [6]These authors contributed equally: Hesham Elsaman, Evgeny Golubtsov. ✉e-mail: wagenerj@tcd.ie

referred to as fungicidal activity[6,7]. We have previously shown that azoles trigger the formation of cell wall patches in *Aspergillus fumigatus*, the most important human pathogen of the *Aspergillus* genus. These patches are essentially cell wall bulges based on excessive synthesis of cell wall carbohydrates which extend into the lumen of the hyphae and which cause cell wall stress and contribute to the fungicidal activity of azoles against this mold[7]. The mechanisms leading to the formation of these cell wall patches, which apparently are not formed in yeasts after azole exposure, remained unknown.

The ergosterol biosynthesis pathways of fungi differ in some parts between species. While the upstream precursors of the biosynthesis (e.g., farnesyl pyrophosphate, squalene, squalene epoxide, or lanosterol) as well as the final product (ergosterol) generally appear to be the same and the biosynthesis is carried out by a highly conserved set of enzymes, the sequence of the enzymatic steps required for synthesis varies depending on the species[8–11]. To form the initial sterols of the pathway, farnesyl pyrophosphate, the last product of the mevalonate pathway, is converted to squalene by squalene synthase (Erg9 in *Saccharomyces cerevisiae* (baker's yeast); Fig. 1A and Supplementary Fig. 1A). Squalene is then converted to squalene epoxide by squalene epoxidase (Erg1 in baker's yeast). Squalene epoxide is converted by lanosterol synthase (Erg7 in baker's yeast) to lanosterol, which is the first sterol of the ergosterol biosynthesis pathway[12].

In the baker's yeast as well as in the pathogenic yeast *Candida albicans*, lanosterol (4,4,14-trimethylcholesta-8,24(25)-dien-3β-ol (**1**); Fig. 1A and Supplementary Fig. 1A) is the substrate of CYP51 (also known as Erg11 in yeasts) which converts it to 4,4-dimethylcholesta-8,14,24(25)-trien-3β-ol (**6**). This product is then further processed by a number of other enzymes to ergosterol (**14**)[11]. However, in the mold *A. fumigatus*, lanosterol (**1**) is converted by sterol C24-methyltransferase

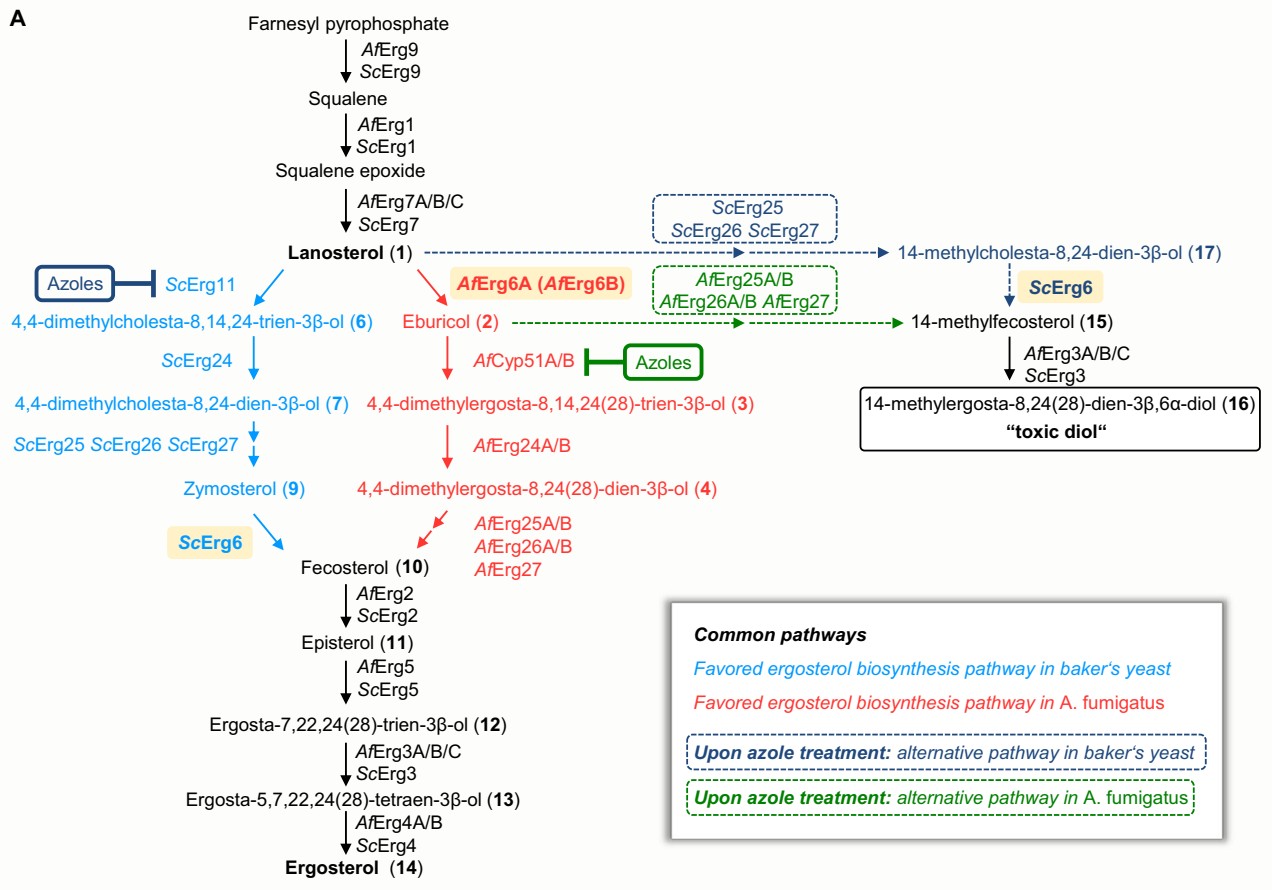

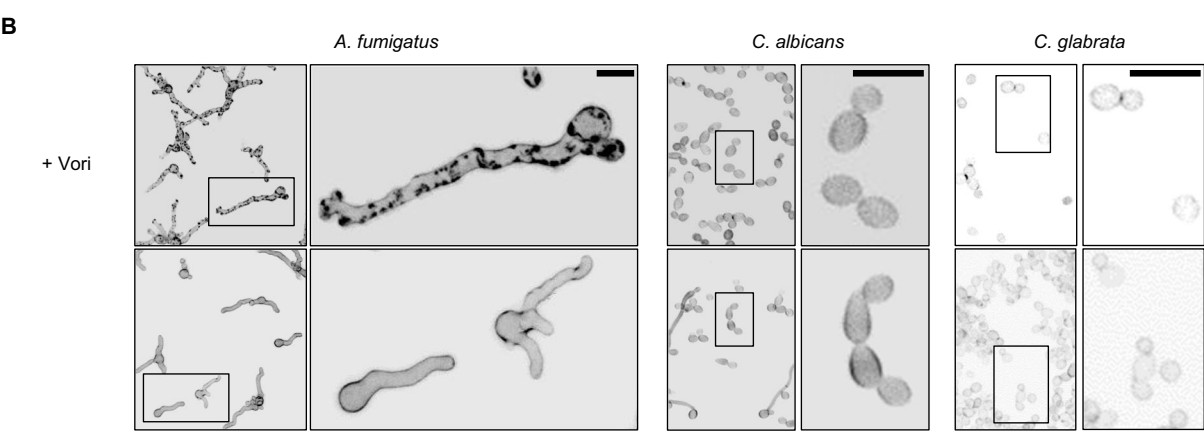

**Fig. 1 | The sterol biosynthesis pathway and the azole-induced formation of cell wall carbohydrate patches depends on the species. A** Ergosterol biosynthesis pathways in the mold *A. fumigatus* (*Af*; right pathway in red) and yeast *S. cerevisiae* (*Sc*; left pathway in light blue). In *A. fumigatus*, C24-methylation of lanosterol (**1**) by sterol C24-methyltransferase (*Af*Erg6A; highlighted in light orange) is favored over C14-demethylation. In *S. cerevisiae*, lanosterol (**1**) is the favored substrate of sterol C14-demethylase (*Sc*Erg11). Both pathways converge with the formation of fecosterol (**10**), which is then further processed into ergosterol (**14**). Upon inhibition of sterol C14-demethylase (AfCyp51A/B and *Sc*Erg11) by azoles, alternative pathways may lead to the formation of a "toxic diol" (dashed arrows; alternative baker's yeast pathway in dark blue, alternative *A. fumigatus* pathway in green). In *S. cerevisiae*, lanosterol (**1**) is converted to 14-methylcholesta-8,24-dien-3β-ol (**17**), which in turn is converted to 14-methylfecosterol (14-methylergosta-8,24(28)-dien-3β-ol) (**15**). In *A. fumigatus*, eburicol (**2**) is directly converted to 14-methylfecosterol (**15**). 14-methylfecosterol (**15**) is then converted to the 14-methylergosta-8,24(28)-dien-3β,6α-diol (**16**) which is considered a "toxic diol". Ergosterol biosynthesis enzymes in *A. fumigatus* (*Af*) and *S. cerevisiae* (*Sc*): squalene synthase (*Af*Erg9, *Sc*Erg9), squalene epoxidase (*Af*Erg1, *Sc*Erg1), lanosterol synthase (*Af*Erg7A/B/C, *Sc*Erg7), sterol C24-methyltransferase (*Af*Erg6A, *Af*Erg6B and *Sc*Erg6; highlighted in light orange), sterol C14-demethylase (*Af*Cyp51A/B, *Sc*Erg11), sterol C14-reductase (*Af*Erg24A/B, *Sc*Erg24), sterol C4-demethylase complex (*Af*Erg25A/B, *Af*Erg26A/B, *Af*Erg27, *Sc*Erg25, *Sc*Erg26, *Sc*Erg27), sterol C8-isomerase (*Af*Erg2, *Sc*Erg2), sterol C22-desturase (*Af*Erg5, *Sc*Erg5), sterol C5-desaturase (*Af*Erg3A/B/C, *Sc*Erg3), sterol C24 reductase (*Af*Erg4A/B, *Sc*Erg4). **B** Conidia of *A. fumigatus* wild type and two *Candida* species (*C. albicans* ATCC14053 and *C. glabrata* ATCC2950) were inoculated in Sabouraud medium and incubated at 37 °C. After 9.5 h of incubation, the samples were either fixed and stored at 4 °C (control) or, after the medium was supplemented with 3 μg ml⁻¹ voriconazole (+Vori), further incubated at 37 °C. After 15 h of additional incubation, the voriconazole-exposed hyphae and yeasts were also fixed. Samples were then stained with calcofluor white and analyzed with a confocal laser scanning microscope. Depicted are representative images of z-stack projections of optical stacks of the calcofluor white fluorescence covering the entire hyphae in focus. Upper panel, voriconazole-treated hyphae; lower panel, controls. Bars represent 10 μm. Data were representative of three experiments.

to eburicol (4,4,14-trimethylergosta-8,24(28)-dien-3β-ol (**2**))[8]. Eburicol (**2**) is then the substrate of CYP51, followed by further processing to ergosterol (**14**)[8]. Consequently, inhibition of CYP51 results in the accumulation of different precursors in the yeasts *S. cerevisiae* and *C. albicans* on the one hand and in the mold *A. fumigatus* on the other hand.

Notably, it is not in the first instance the lack of ergosterol that is responsible for the antifungal effects of azoles on baker's yeast and *Candida* species. Instead, the accumulation of toxic sterols comes into play[13]. This is best illustrated by the fact that inactivation of sterol C5-desaturase (Erg3 in baker's yeast) causes azole resistance in *C. albicans*, *Candida glabrata*, and *S. cerevisiae*[14–17]: Upon inhibition of CYP51, lanosterol (**1**) accumulates, which is converted to 14-methylfecosterol (14-methylergosta-8,24(28)-dien-3β-ol (**15**); Fig. 1 and Supplementary Fig. 1B). Erg3 then converts it to 14-methylergosta-8,24(28)-dien-3β,6α-diol (**16**) which is considered the "toxic diol"[18,19]. Azole-treated yeast strains that lack the enzymatic function of Erg3 do not form the "toxic diol" (**16**). Instead, 14-methylfecosterol (**15**) accumulates, allowing them to grow (Fig. 1 and Supplementary Fig. 1B).

We hypothesized that the differences in the ergosterol biosynthesis pathways of yeasts and molds are causally involved in the different effects of azole antifungals against these fungi. To investigate the molecular nature of the different modes of action, we constructed *A. fumigatus* ergosterol biosynthesis mutants which allowed us to investigate the importance of individual ergosterol biosynthesis enzymes and dissect and assign the antimicrobial potential of individual steroid intermediates and derivatives. Our results show that it is primarily the accumulation of the ergosterol biosynthesis intermediate eburicol that has detrimental effects on *A. fumigatus*'s cell physiology and is responsible for the strong antimicrobial activity of azole antifungals against this pathogen.

## Results

### Erg6A but not Erg6B is essential in *A. fumigatus*

Azoles induced the formation of cell wall carbohydrate patches in *A. fumigatus*, but not in pathogenic yeasts such as *C. albicans* and *C. glabrata* (Fig. 1B). We speculated that these differences might be linked with the species-specific differences in the sequence of the enzymatic steps involved in ergosterol biosynthesis (Fig. 1A).

In the first step, we aimed to generate a mutant in which the function of the sterol C24-methyltransferase (Erg6 in baker's yeast) can be repressed. The genome of *A. fumigatus* harbors two genes that encode Erg6 homologs, AFUA_4G03630 and AFUA_4G09190, which we named *erg6A* and *erg6B*, respectively. Since we initially assumed that the two Erg6 homologs might be functionally redundant, we planned to generate a double mutant in which one of the genes (*erg6B*)

was deleted, and the expression of the other gene (*erg6A*) can be downregulated by replacing the endogenous promoter with a doxycycline-inducible Tet-On promoter[20,21]. However, we found that repression of *erg6A* alone prevents the growth of *A. fumigatus*, while deletion of *erg6B* did not result in any apparent growth phenotype (Fig. 2A). This suggested that the sterol C24-methyltransferase step, in contrast to its role in yeast[22], is essential for the viability of *A. fumigatus* and that *erg6B* cannot compensate for the lack of Erg6A. When comparing the growth of the conditional *erg6A*$_{tetOn}$ mutant with that of a similarly constructed conditional CYP51 *A. fumigatus* mutant (*cyp51A*$_{tetOn}$ Δ*cyp51B*)[7], we found that the macroscopically visible growth impairment occurred at similarly reduced doxycycline concentrations (Fig. 2B).

### Depletion of CYP51 but not of Erg6A triggers cell wall patch formation

We next analyzed whether repression of *erg6A* results in the formation of cell wall carbohydrate patches, similar to what we had previously found it in *A. fumigatus* that was treated with azoles or where CYP51 was depleted[7]. As shown in Fig. 2C, depletion of CYP51 resulted in the accumulation of chitin-rich calcofluor white-stainable cell wall patches. In contrast, depletion of Erg6A did not result in the accumulation of cell wall patches (Fig. 2C). Notably, compared to hyphae of the conditional CYP51 mutant, the hyphae of the conditional *erg6A*$_{tetOn}$ mutant continued to grow for much longer after the removal of doxycycline. We speculated that the time of survival of the *erg6A*$_{tetOn}$ mutant after doxycycline depletion also differs from that of the CYP51 mutant after doxycycline depletion. This could potentially correlate with a delay in the formation of cell wall patches. To study this, we used an assay which is based on the evaluation of mitochondrial dynamics to assess the viability of *Aspergillus* hyphae over time[7]. As shown in Fig. 2D, the half-life of *erg6A*$_{tetOn}$ mutant hyphae after doxycycline depletion was approximately double that of the CYP51 mutant hyphae after doxycycline depletion. Since the photos depicted in Fig. 2C were taken after a similar incubation time (40 h) under similar experimental conditions, this indicates that the *erg6A*$_{tetOn}$ mutant died without forming cell wall patches prior to death.

### Depletion of Erg6A increases azole tolerance of *A. fumigatus*

Our results demonstrated that Erg6A is essential for the viability of *A. fumigatus*. But our results also suggested that the cell biological nature of Erg6A depletion-induced death is somewhat different from that of the CYP51 depletion- or inhibition-induced death because (1) death occurs much slower and (2) no cell wall patches get formed prior to death. We speculated that the accumulation of different ergosterol precursors could be linked with this. Erg6A depletion would

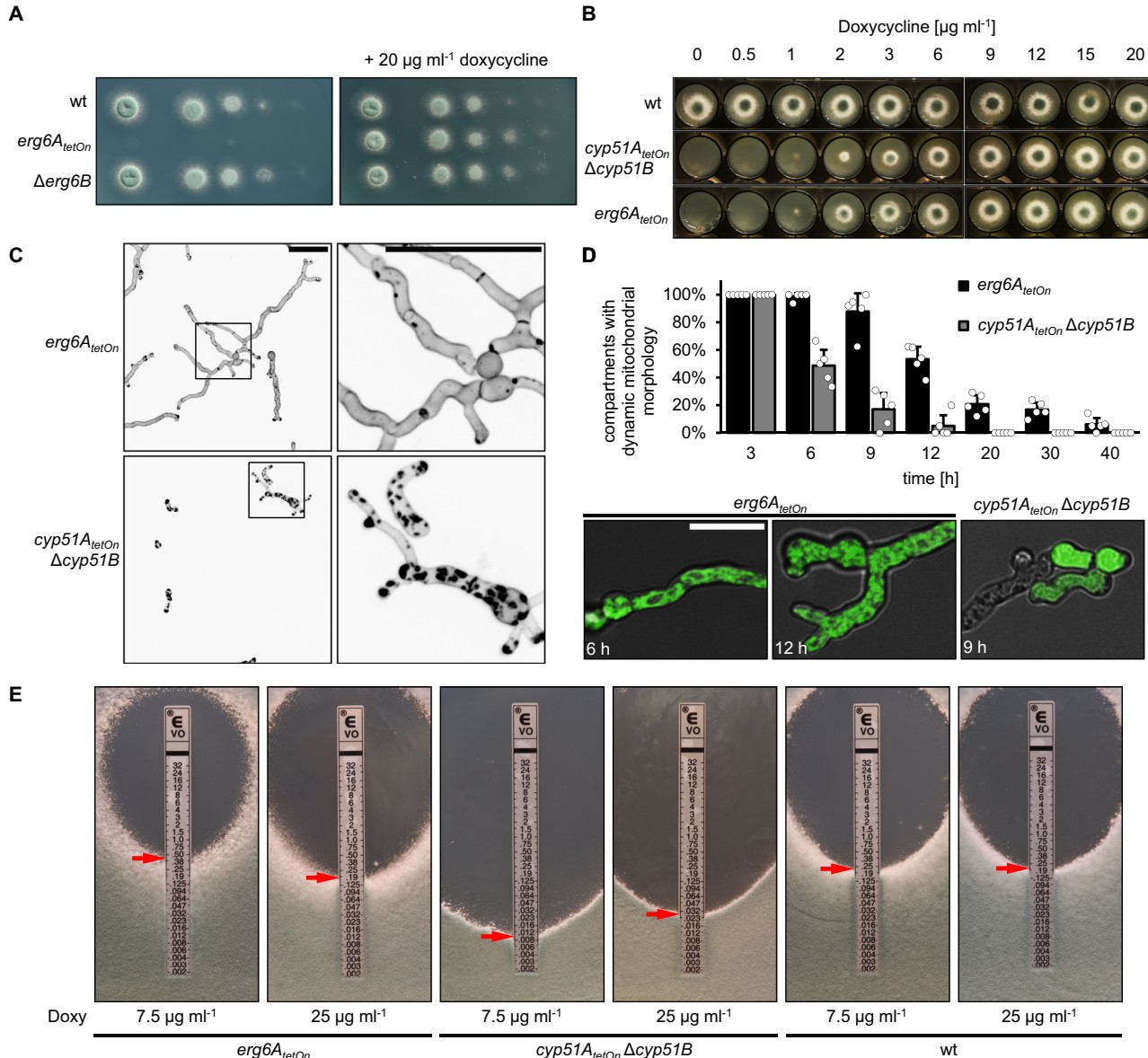

**Fig. 2 | The impact of sterol C24-methyltransferase depletion on viability, cell wall patch formation, and azole susceptibility. A** In a series of tenfold dilutions derived from a starting suspension of $5 \times 10^7$ conidia ml$^{-1}$ of wild type (wt), the conditional mutant *erg6A*$_{tetOn}$, and the deletion mutant Δ*erg6B*, aliquots of 3 μl were spotted onto AMM agar plates. AMM was supplemented with 20 μg ml$^{-1}$ doxycycline when indicated. Agar plates were incubated at 37 °C, representative photos were taken after 28 h. **B** $1.5 \times 10^3$ conidia of the indicated strains were spotted on AMM agar supplemented with the indicated amount of doxycycline. The plates were then incubated at 37 °C, representative photos were taken after 40 h. **C, D** Conidia of the *erg6A*$_{tetOn}$ and *cyp51A*$_{tetOn}$ Δ*cyp51B* mutants which express mitochondria-targeted GFP (**D**) or not (**C**) were inoculated in Sabouraud medium supplemented with 20 μg ml$^{-1}$ doxycycline and incubated at 37 °C. After 6 h, doxycycline was depleted by washing the wells three times with Sabouraud medium without doxycycline. The hyphae were then incubated in Sabouraud medium without doxycycline at 37 °C for an additional 40 h. **C** Hyphae were stained with calcofluor white and analyzed with a confocal microscope. Depicted are representative images of z-stack projections of optical stacks of the calcofluor white fluorescence covering the entire hyphae in focus. The right images show magnifications of the framed sections in the left images. Bars represent 50 μm and are applicable to all images in the respective panel. Data were representative of five independent experiments. (**D**, column graph) At the indicated time points after doxycycline depletion, the viability of the hyphae of the indicated strains expressing mitochondria-targeted GFP was analyzed with time-lapse spinning disc confocal microscopy. The bars indicate the percentage of hyphal compartments with evident mitochondrial dynamics (viable compartments). Each bar represents the mean of five technical replicates (data points) for each timepoint, with an average of ~80 analyzed compartments per strain for each timepoint. The error bars indicate standard deviations based on the five technical replicates. (**D**, lower panel) Representative overlay images of the bright-field channel and of z-stack projections of optical stacks of the GFP fluorescence covering the entire hyphae in focus. Depicted are exemplary images of a viable hypha with tubular mitochondrial morphology (also showing mitochondrial dynamics in time-lapse microscopy; left image) and of hyphae with fragmented mitochondria (showing no mitochondrial dynamics in time-lapse microscopy; middle image) or with no or cytosolic GFP signal (right image) which were considered to be dead. Bars represent 50 μm and are applicable to all respective subpanels. **E** $1 \times 10^6$ conidia of the indicated strains were spread on Sabouraud agar plates. Agar was supplemented with the indicated amount of doxycycline (Doxy) to achieve a different induction of the conditional promoters. Voriconazole Etest strips were applied. The plates were incubated at 37 °C and representative photos were taken after 42 h.

presumably result in the accumulation of lanosterol (**1**), CYP51 depletion results in the accumulation of eburicol (**2**) (Fig. 1A), according to the literature[23]. This would be compatible with a model where the accumulation of eburicol is more toxic than the accumulation of lanosterol.

We then tested whether Erg6A depletion impacts on the azole susceptibility of *A. fumigatus*, and compared this to the effects of depletion of CYP51 on azole sensitivity. Reduced expression of CYP51 resulted in increased susceptibility to azoles (Fig. 2E), which is perfectly in line with overexpression of CYP51 having the opposite effect and being a major resistance mechanism against azoles in *A. fumigatus*[24]. Interestingly, culturing the conditional *erg6A$_{tetOn}$* mutant under less inducing conditions resulted in reduced susceptibility to azoles when compared to wild type and the *erg6A$_{tetOn}$* mutant under more inducing conditions (Fig. 2E and Supplementary Fig. 2). This indicates that a reduced synthesis of eburicol is beneficial for *A. fumigatus* if CYP51 is inhibited by azoles at the same time.

## Erg1 and Erg9 are essential, but their depletion does not trigger patch formation

To confirm that the azole-induced cell wall carbohydrate patches are not just a result of reduced sterol biosynthesis, we constructed conditional squalene synthase (Erg9) and squalene epoxidase (Erg1) mutants[25]. Erg9 and Erg1 are both essentially required to form squalene or, respectively, squalene epoxide which are precursors of lanosterol, the first sterol in the ergosterol biosynthesis pathway[12] (Fig. 1A). Consequently, downregulation of *erg9* or *erg1* should result in disruption of the whole sterol biosynthesis. The conditional *erg9$_{tetOn}$* and *erg1$_{tetOn}$* mutants were not viable under repressed conditions (Fig. 3A)[25]. Interestingly, none of these mutants formed cell wall patches under repressed conditions (Fig. 3B). Similarly, *A. fumigatus* wild type, which expresses mitochondria-targeted green fluorescent protein (GFP) as viability marker, did not form cell wall patches when treated with the Erg1 inhibitor terbinafine (Fig. 3C). The mitochondrial dynamics-based viability marker demonstrated that the terbinafine-treated hyphae had died without forming patches (Fig. 3C). This demonstrates that disruption of the whole sterol biosynthesis is lethal, but does not result in the formation of cell wall patches.

Next, we analyzed the impact of reduced *erg9* and *erg1* expression on azole susceptibility. As shown in Fig. 3D and Supplementary Fig. 3, less inducing conditions resulted in slightly increased azole susceptibility of the conditional *erg9$_{tetOn}$* and *erg1$_{tetOn}$* mutants compared to wild type. This is consistent with the results of previous studies showing that azole antifungals and the Erg1 inhibitor terbinafine act synergistically on *A. fumigatus*[26,27]. In contrast, more inducing conditions resulted in a similar to minimally reduced azole susceptibility compared to wild type (Fig. 3D and Supplementary Fig. 3). Taken together, these findings suggest that azoles are more effective when the total sterol biosynthesis is downregulated.

## Impact of Erg9, Erg6A, and CYP51 depletion on fungal sterol patterns

The conclusions presented above were based on the assumption that in *A. fumigatus,* depletion of CYP51, similar to treatment with azole antifungals, would result in accumulation of eburicol (**2**), and depletion of Erg6A would result in accumulation of lanosterol (**1**). Furthermore, we would not expect a drastic change in the sterol pattern upon depletion of Erg9 or Erg1 because any synthesized lanosterol could still be readily converted to ergosterol (**14**). To confirm these suppositions, we analyzed the sterol patterns of the conditional CYP51, *erg6A$_{tetOn}$*, and *erg9$_{tetOn}$* mutants under inducing as well as repressing conditions (Fig. 4A and Supplementary Tables 1, 2). Different concentrations of doxycycline, the chemical used to induce the conditional TetOn promoters, did not significantly alter the sterol pattern of the wild type (Supplementary Fig. 4A). When compared with the wild type,

depletion of Erg9 resulted in a slight but significant decrease in the amount of ergosterol relative to other sterols (Fig. 4B and Supplementary Fig. 4B). Further depletion of Erg9 (by further reducing the doxycycline concentration) did not result in a remarkable alteration of this pattern (Fig. 4B and Supplementary Fig. 4B). Depletion of Erg6 resulted in a significant decrease of ergosterol and a relative increase of lanosterol (Fig. 4B and Supplementary Fig. 4B). Depletion of CYP51 resulted in a dramatic increase of eburicol (**2**), but also in a remarkable increase of the presumably "toxic diol" (14-methylergosta-8,24(28)-dien-3β,6α-diol (**16**)), and lanosterol (**1**). The impact of low doxycycline concentrations on the sterol patterns was much more pronounced in the conditional CYP51 mutant compared to the conditional *erg6A$_{tetOn}$* mutant (Supplementary Fig. 4B). We speculated that this could be linked with the complete lack of *cyp51B* in the CYP51 mutant (*cyp51A$_{tetOn}$* Δ*cyp51B*) and the inability of the Tet-On promoter-dependent *cyp51A* to compensate for this (see also increased azole susceptibility under fully induced conditions; Fig. 2E).

These findings confirmed our expectation that Erg6 depletion results in the accumulation of lanosterol. They also confirmed the previously reported accumulation of eburicol upon depletion of CYP51[23,28]. However, our findings also demonstrate that the presumably "toxic diol" (**16**) is formed in *A. fumigatus* when CYP51 functionality is lacking, which is in agreement with a previous report[29].

## ERG3-depleted hyphae form cell wall patches when exposed to azoles

Depletion of CYP51 resulted in a significant increase of the presumably "toxic diol" (**16**). Accumulation of this sterol is responsible for the antifungal effect of azole antifungals against *S. cerevisiae* and *Candida* species[13]. Accordingly, its accumulation might also explain the antifungal effects of azoles on *A. fumigatus*, e.g., the formation of cell wall carbohydrate patches. The formation of the "toxic diol" in yeasts requires the incomplete (defective) reaction of the sterol C5-desaturase. If this model also holds true in *Aspergillus*, disruption of the desaturation step of 14-methylfecosterol (**15**) by inactivating sterol C5-desaturase should cause azole resistance in *A. fumigatus*, similar as it was previously shown for *S. cerevisiae* and *Candida* species[14,15,17]. The genome of *A. fumigatus* harbors three genes that encode homologes of *S. cerevisiae* Erg3, AFUA_6G05140 (*erg3A*), AFUA_2B00320 (*erg3B*), and AFUA_8G01070 (*erg3C*). While mutants lacking the individual enzymes as well as an *erg3A erg3B* double mutant have been reported and characterized previously[8,30], no mutants that lack the other combinations (*erg3A erg3C* and *erg3B erg3C*) or all three enzymes have been reported. To characterize the roles of Erg3A, Erg3B, and Erg3C, we constructed single, double, and triple mutants (Fig. 5A, B). As shown in Fig. 5B, the lack of individual Erg3 enzymes, as well as the lack of Erg3A and Erg3B or Erg3B and Erg3C, did not result in any significant impact on growth. In contrast, the lack of Erg3A and Erg3C, as well as the lack of Erg3A, Erg3B, and Erg3C resulted in reduced growth rates and reduced production of asexual spores (conidia) (Fig. 5B).

We then asked whether the conditional ERG3 triple mutant (*erg3A$_{tetOn}$* Δ*erg3B* Δ*erg3C*) still forms the presumably "toxic diol" upon exposure to azoles. To this end, we analyzed the sterol profiles of the repressed conditional ERG3 mutant in the absence and presence of azoles and compared them with the sterol profiles of wildtype cultured under similar conditions (Fig. 5C, D and Supplementary Table 3). As expected, the ERG3 mutant under repressed conditions did not form ergosterol (**14**) because of the lack of sterol C5-desaturase function, but instead accumulated Δ7-sterols summed as "other sterols" (relative percentages of total sterols: 74% ergosta-7,22,24(28)-trien-3β-ol (**12**), 15% ergosta,7,24(28)-dien-3β-ol (episterol, **11**), 9% ergosta-7,22-dien-3β-ol (5-dihydroergosterol); Fig. 5D and Supplementary Table 3). The azole-treated wild type accumulated eburicol, lanosterol and the presumably "toxic diol" with a concomitant relative decrease of ergosterol

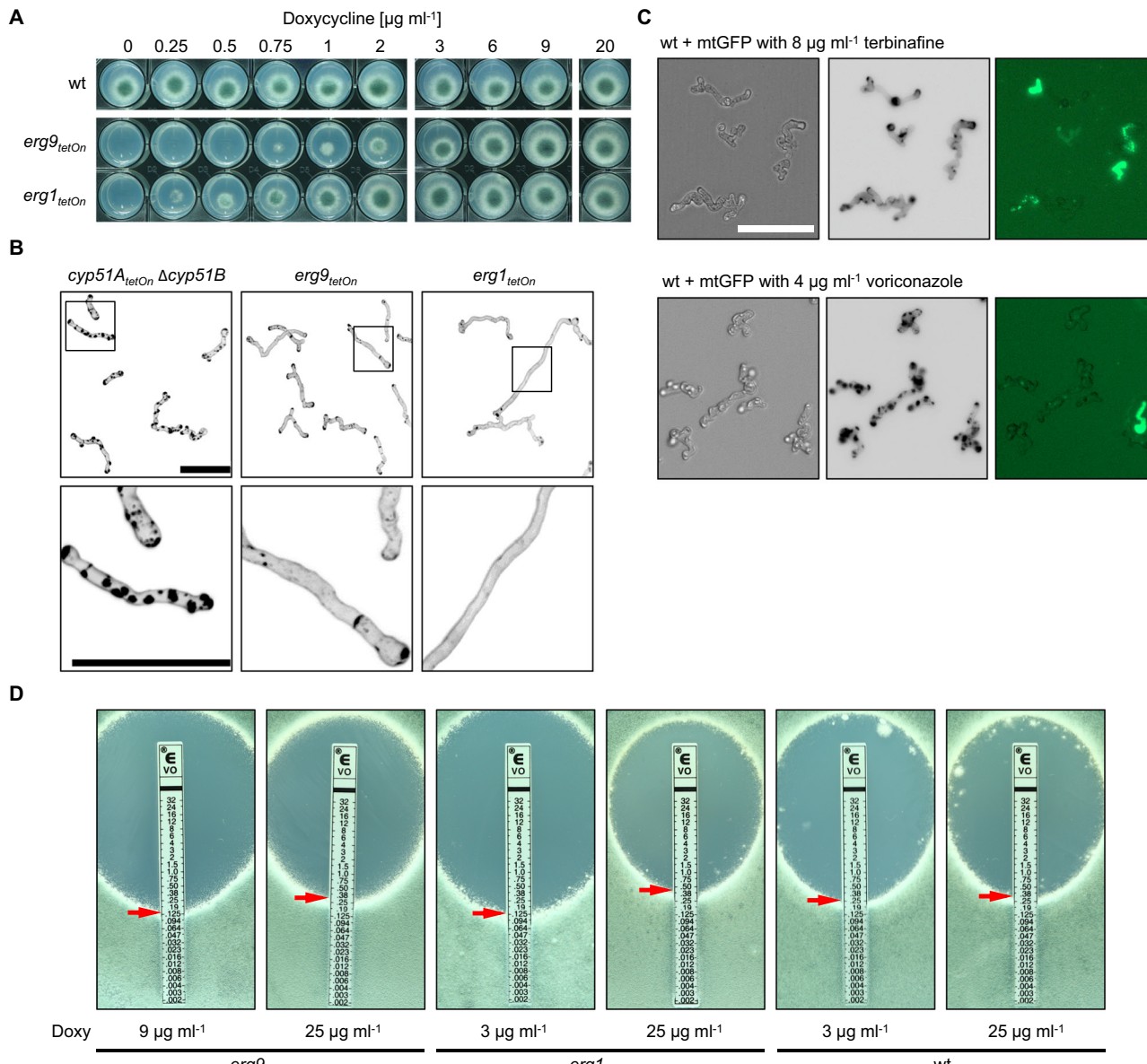

**Fig. 3 | Depletion of squalene synthase or squalene epoxidase does not trigger excessive cell wall patch formation, but their altered expression results in changes in azole sensitivity. A** $1.5 \times 10^3$ conidia of wild type (wt) or of the conditional squalene synthase ($erg9_{tetOn}$) or of the squalene epoxidase ($erg1_{tetOn}$) mutants were spotted on AMM agar supplemented with the indicated amount of doxycycline. The plates were then incubated at 37 °C, representative photos were taken after 40 h. **B** Conidia of the indicated strains were inoculated in Sabouraud medium supplemented with 20 µg ml⁻¹ doxycycline and incubated at 37 °C. After 6 h, doxycycline was depleted by washing the wells three times with Sabouraud medium without doxycycline. The hyphae were then incubated in Sabouraud medium without doxycycline at 37 °C for an additional 40 h and subsequently stained with calcofluor white and analyzed with a confocal microscope. Depicted are representative images of z-stack projections of optical stacks of the calcofluor white fluorescence covering the entire hyphae in focus. The lower images show magnifications of the framed sections in the upper images. Bars represent 50 µm and are applicable to all images in the respective panel. Data were representative of four

independent experiments. **C** Conidia of wildtype expressing mitochondria-targeted GFP were inoculated in Sabouraud medium. After 8 h incubation at 37 °C, medium supplemented with 8 µg ml⁻¹ terbinafine (upper panel) or with 4 µg ml⁻¹ voriconazole (lower panel). After an additional 16 h incubation at 37 °C, hyphae were analyzed with a fluorescence microscope. Fluorescence signals were analyzed sequentially, first the GFP signal was recorded followed by staining with calcofluor white and recording of the calcofluor white signal. Depicted are representative images of bright-field (left) and z-stack projections of optical stacks of the calcofluor white fluorescence (middle) and GFP fluorescence (right) after deconvolution, that cover the entire hyphae in focus. The bar represents 50 µm and is applicable to all images. Data were representative of four independent experiments. **D** $1 \times 10^6$ conidia of the indicated strains were spread on Sabouraud agar plates. Agar was supplemented with the indicated amount of doxycycline (Doxy) to achieve a different induction of the conditional promoters. Voriconazole Etest strips were applied. The plates were incubated at 37 °C and representative photos were taken after 42 h.

(Fig. 5D). While the azole-exposed ERG3 mutant under repressed conditions also accumulated eburicol, lanosterol and other sterols (relative percentages of total sterols: 66% ergosta-7,22,24(28)-trien-3β-ol (**12**), 10% ergosta-7,22-dien-3β-ol (5-dihydroergosterol), 8% ergosta,7,24(28)-dien-3β-ol (episterol, **11**)), it did not form the presumably toxic diol (Fig. 5D and Supplementary Table 3). This confirms

that the Erg3 enzymes in *A. fumigatus* are needed to form 14-methyl-lergosta-8,24(28)-dien-3β,6α-diol ("toxic diol" (**16**)) upon exposure to azoles. Interestingly, only traces of 14-methylfecosterol (14-methyl-lergosta-8,24(28)-dien-3β-ol) (**15**) were detected both in wild type and the ERG3 mutant under repressed conditions after azole exposure (0.36 and 0.26%, respectively; Supplementary Table 3).

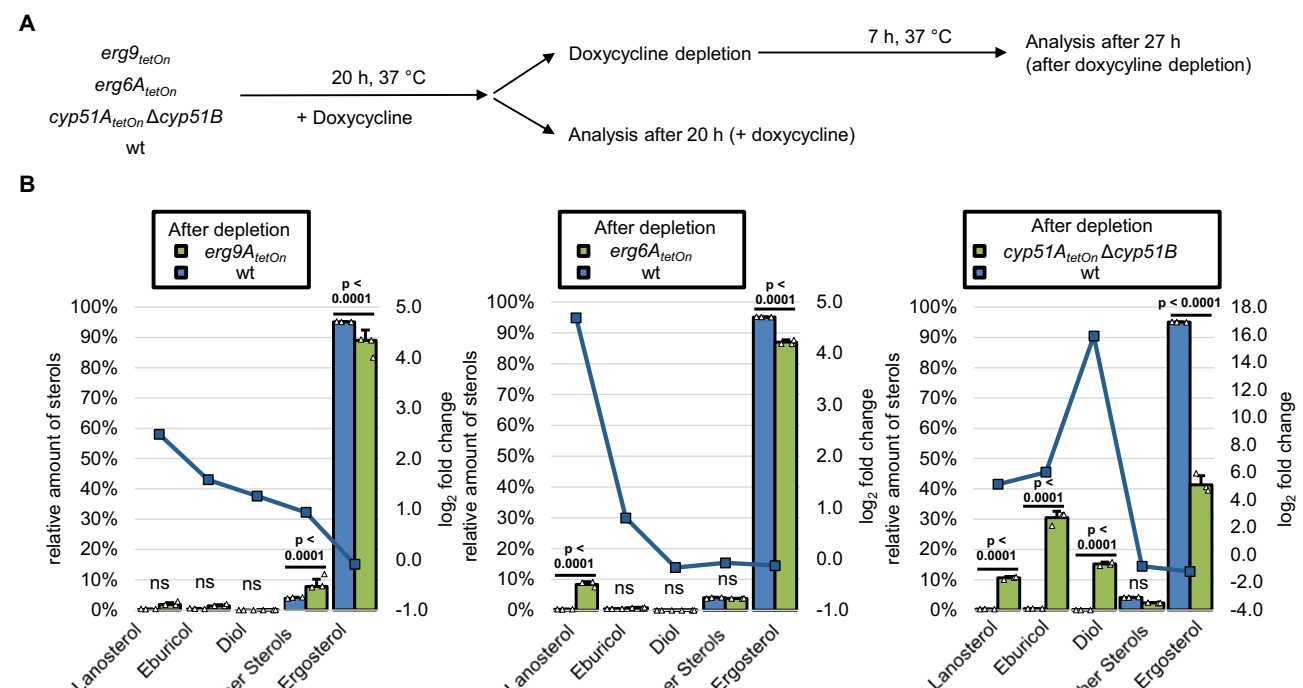

**Fig. 4 | Sterol profiles of squalene synthase-, sterol C24-methyltransferase- and sterol C14-demethylase-depleted *A. fumigatus* hyphae. A** Conidia of wild type (wt) and the indicated mutants were inoculated in a Sabouraud liquid medium. Cultures with the conditional sterol C24-methyltransferase mutant (*erg6A_{tetOn}*), the conditional squalene synthase (*erg9_{tetOn}*), and the respective wild-type control (wt) were supplemented with 2 µg ml⁻¹ doxycycline. Cultures with the conditional sterol C14-demethylase mutant (*cypS1A_{tetOn} ΔcypS1B*) and the respective wild-type control (wt) were supplemented with 3 µg ml⁻¹ doxycycline. After 20 h incubation in a rotary shake at 37 °C, mycelium was either directly harvested or washed with and transferred into Sabouraud liquid medium without doxycycline, incubated in a rotary shake at 37 °C for another 7 h, and then harvested. For each condition, three biological replicates were cultured. **B** The sterol patterns of the harvested mycelia were analyzed by gas chromatography-mass spectrometry (GC-MS). The column graphs show the relative amounts (percentage of total sterol, left y-axis) of the indicated sterols for the indicated strains before and after doxycycline depletion. The data points with the square symbols indicate the log₂-fold change (right y-axis) in the amount of the respective sterol of the pairwise comparison of the conditions shown in the individual graphs. The log₂-fold change data points were connected by lines for a better visual illustration of the changes in the profiles. Each column bar represents the mean of three replicates (data points) per condition, the error bars indicate standard deviations. Data were representative of five (*erg6A_{tetOn}*; *cypS1A_{tetOn} ΔcypS1B*) and one (*erg9_{tetOn}*) experiments conducted under similar conditions. Statistical significance was set at $p < 0.05$, and calculated with a two-way ANOVA with Tukey's multiple comparison test. $p$ values are indicated in the graphs; ns not significant. Data were representative of this experiment are shown in Supplementary Fig. 4. Source data are provided as a Source Data file.

We next evaluated whether the formation of the "toxic diol" (**16**) is required for the formation of the azole-induced cell wall patches. To this end, we exposed hyphae of the ERG3 mutant under repressed conditions to azoles. As shown in Fig. 6A, the lack of ERG3 did not suppress the formation of cell wall patches after exposure to azole antifungals. This demonstrates that it is not the accumulation of the "toxic diol" which triggers the formation of the azole-induced cell wall patches in *A. fumigatus*.

### Depletion of ERG3 decreases azole tolerance of *A. fumigatus*
In *S. cerevisiae* and *Candida* species, the lack of Erg3 results in azole resistance[13,18,19]. We asked whether this also holds true in *A. fumigatus*. Growth tests were performed to analyze the azole susceptibilities of the conditional Erg3 double mutants and the conditional ERG3 triple mutant under induced and repressed conditions. As shown in Fig. 6B, the azole susceptibilities of the *erg3A_{tetOn} Δerg3B* mutant and the *erg3A_{tetOn} Δerg3C* mutant were not significantly changed under induced or repressed conditions when compared with wild type. Similarly, the ERG3 triple mutant under induced, effectively representing a mutant lacking Erg3B and Erg3C, showed no significantly altered azole susceptibility (Fig. 6B and Supplementary Fig. 5). However, under repressed conditions, the ERG3 triple mutant was significantly more susceptible to azoles (Fig. 6B and Supplementary Fig. 5). This shows that Erg3A/B/C are at least partially functionally redundant and counteract the toxic effects of azole antifungals in

*A. fumigatus*. This is the exact opposite of what has been described for the mode of action of azoles and the role of Erg3 therein for baker's yeast and *Candida* species.

### Mitochondrial complex III dysfunction enhances conversion of eburicol to 14-methylergosta-8,24(28)-dien-3β,6α-diol
We have previously shown that mutants that lack a functional mitochondrial complex III show an unusual azole susceptibility phenotype[7,31]. Lack of a functional complex III results in a significantly reduced minimal inhibitory concentration (MIC) of azole antifungals. At the same time, the complex III mutants were able to survive at azole concentrations above the MIC, which is in marked contrast to the wild type which is effectively killed. This correlates with a decrease in azole-induced cell wall patches[7]. In our earlier work, however, the reason for the reduced fungicidality remained unexplained. A possible explanation for the unusual sensitivity to azoles could be related to an altered processing of sterols. To explore this, the sterol profile of a conditional cytochrome *c* mutant (*cycA_{tetOn}*) and a conditional Rieske protein mutant (*rip1_{tetOn}*) were analyzed and compared to the wild type (Fig. 7A and Supplementary Table 4). Under repressed conditions, the sterol profiles of the *cycA_{tetOn}* and *rip1_{tetOn}* mutants were similar to that of wild type (Fig. 7B). After exposure to azoles, the ergosterol levels decreased more rapidly in the *cycA_{tetOn}* and *rip1_{tetOn}* mutant than in the wild type (Fig. 7C). Surprisingly, significantly more eburicol accumulated in the azole-treated wild type than in the azole-treated repressed

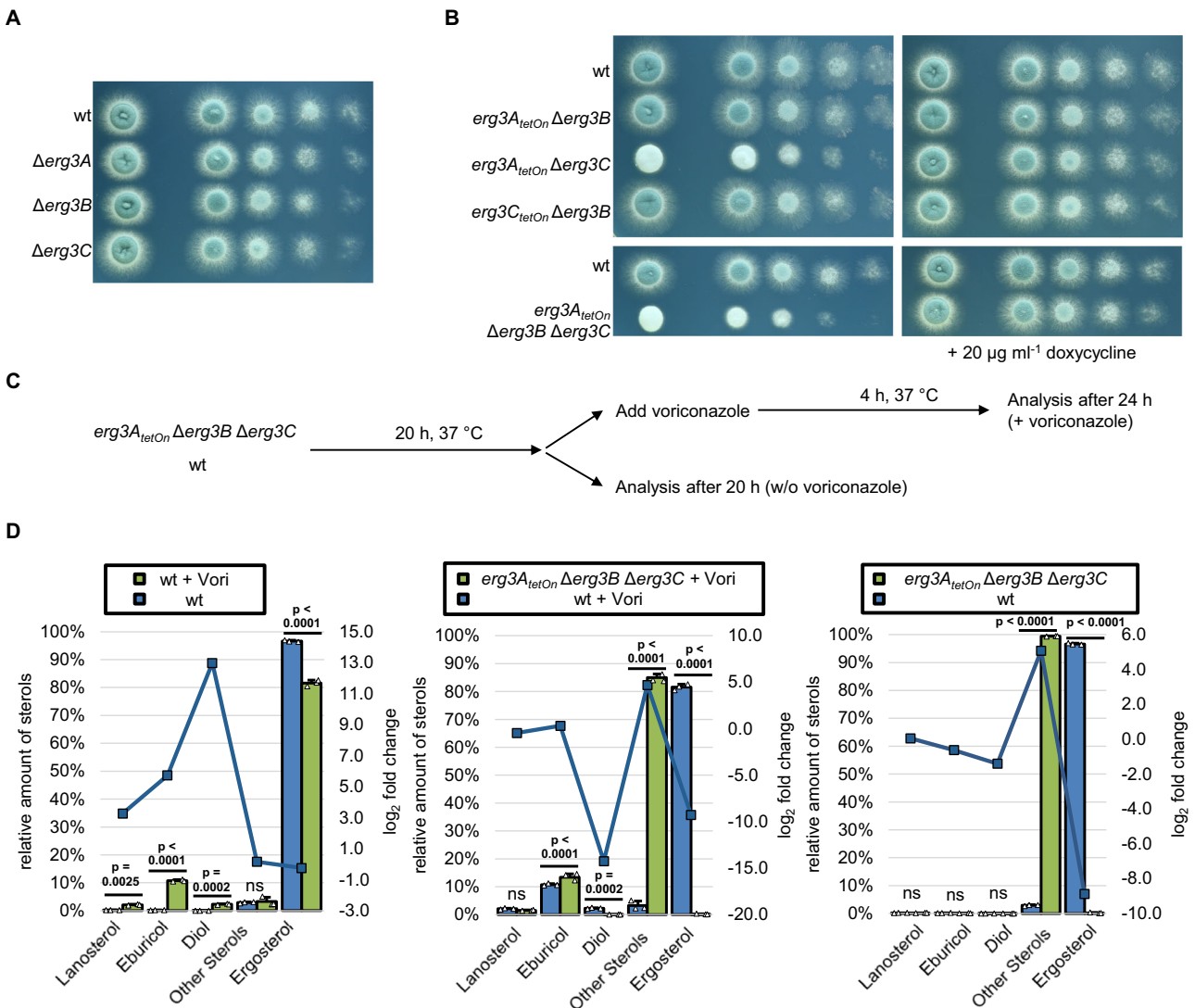

**Fig. 5 | Construction of a conditional sterol C5-desaturase mutant repression and the impact of sterol C5-desaturase depletion on the sterol pattern of azole-treated hyphae. A, B** In a series of tenfold dilutions derived from a starting suspension of $5 \times 10^7$ conidia ml$^{-1}$ of wild type (wt) and the indicated sterol C5-desaturase mutant single, double, and triple mutants, aliquots of 3 μl were spotted onto AMM agar plates. AMM was supplemented with 20 μg ml$^{-1}$ doxycycline when indicated. Agar plates were incubated at 37 °C, representative photos were taken after 28 h. **C, D** Conidia of wild type and the conditional sterol C5-desaturase mutant (erg3A$_{tetOn}$ Δerg3B Δerg3C) were inoculated in the Sabouraud medium. After 20 h incubation in a rotary shake at 37 °C, mycelium was either directly harvested or supplemented with 2 μg ml$^{-1}$ voriconazole (+Vori), incubated in a rotary shake at 37 °C for another 4 h, and then harvested. For each condition, three biological replicates were cultured. **D** The sterol patterns of the harvested mycelia were

analyzed by gas chromatography-mass spectrometry (GC-MS). The column graphs show the relative amounts (percentage of total sterol, left y-axis) of the indicated sterols for the indicated strains under the indicated conditions. The data points with the square symbols indicate the log$_2$-fold change (right y-axis) in the amount of the respective sterol of the pairwise comparison of the conditions shown in the individual graphs. The log$_2$-fold change data points were connected by lines for a better visual illustration of the changes in the profiles. Each column bar represents the mean of three replicates (data points) per condition, the error bars indicate standard deviations. Data were representative of three experiments conducted with two independent conditional ERG3 triple mutants under similar conditions. Statistical significance was set at $p < 0.05$, and calculated with a two-way ANOVA with Tukey's multiple comparison test. $p$ values are indicated in the graphs; ns not significant. Source data are provided as a Source Data file.

---

cycA$_{tetOn}$ and rip1$_{tetOn}$ mutants, whereas significantly more "toxic diol" (14-methylergosta-8,24(28)-dien-3β,6α-diol (**16**)) accumulated in the cycA$_{tetOn}$ and rip1$_{tetOn}$ mutants under repressed conditions than in the wild type (Fig. 7C). This demonstrates that mutants that lack mitochondrial complex III functionality convert eburicol much more efficiently into the diol than the wild type.

## Discussion

Azole antifungals are the mainstay for the treatment and prophylaxis of fungal infections. In yeasts such as *C. albicans* and *C. glabrata*, azole treatment results in the accumulation of lanosterol (**1**), which is converted to 14-methylergosta-8,24(28)-dien-3β,6α-diol (**16**), the so-called

"toxic diol", whose buildup is responsible for the antifungal effect[13]. However, our present study demonstrates that treatment of *A. fumigatus* with azoles results in a predominant accumulation of eburicol (**2**), but also of lanosterol and the "toxic diol", which is in agreement with previous studies[23,28]. The impact of these differences remained unknown.

While the azole activity against yeasts is primarily fungistatic, this drug class exerts a fungicidal effect against molds[6,7]. We have previously shown that azole-induced cell wall carbohydrate patches contribute to the fungicidal effect of azoles against *A. fumigatus*[7]. Based on these findings, we speculated that the eburicol accumulation could contribute to the formation of cell wall carbohydrate patches.

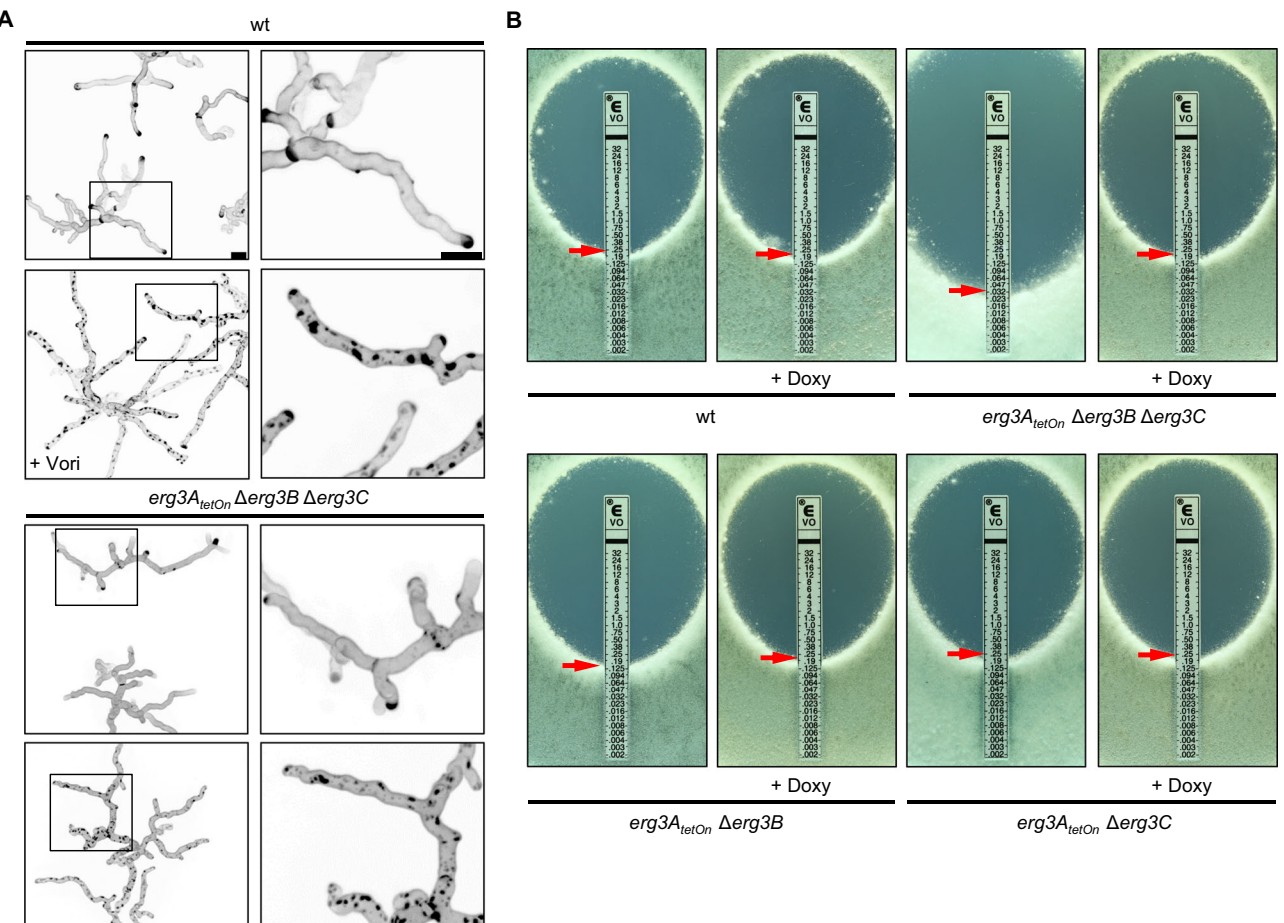

**Fig. 6 | The influence of the sterol C5-desaturase on azole susceptibility and azole-induced cell wall patch formation. A** Conidia of wild type and the conditional sterol C5-desaturase mutant (*erg3A_{tetOn} Δerg3B Δerg3C*) were inoculated in the Sabouraud medium. After 10 h incubation at 37 °C, the samples were either directly analyzed (controls without azole) or the medium was supplemented with 4 µg ml$^{-1}$ voriconazole (+Vori). The voriconazole-exposed hyphae were then incubated at 37 °C for another 5 h. The untreated hyphae (upper images per indicated strain) and the voriconazole-treated hyphae (lower images per indicated strain) were stained with calcofluor white and analyzed with a confocal microscope.

Depicted are representative images of z-stack projections of optical stacks of the calcofluor white fluorescence covering the entire hyphae in focus. The right panels show magnifications of the framed sections in the left panels. Bars represent 50 µm and are applicable to all images in the respective panel. Data were representative of five independent experiments. **B** 1 × 10$^6$ conidia of wild type (wt) and the indicated strains were spread on Sabouraud agar plates. When indicated, agar was supplemented with 25 µg ml$^{-1}$ doxycycline (+Doxy). Voriconazole Etest strips were applied. The plates were incubated at 37 °C and representative photos were taken after 42 h.

Remarkably, in the present study, we were able to show that the lack of other ergosterol biosynthesis enzymes, such as sterol C24-methyltransferase (Erg6A), squalene synthase (Erg9), and squalene epoxidase (Erg1), which are also essential for the viability of *A. fumigatus*, did not trigger the formation of cell wall carbohydrate patches. This demonstrates that the cell wall carbohydrate patch formation is specifically linked with the inhibition of sterol C14-demethylase (CYP51) and the accumulation of eburicol, which is in full agreement with our hypothesis. Based on our previous finding that inhibition of the cell wall patch formation delays azole-induced fungal death[7], our present results support a model where especially the buildup of eburicol is strikingly toxic for *A. fumigatus* (Fig. 7D). This model is further supported by our finding that lowered expression of Erg6A results in increased azole resistance. Mechanistically, this means that repression of *erg6A* leads to reduced conversion of lanosterol to eburicol, and therefore less CYP51 is needed to diminish accumulating eburicol.

But why did the lowered expression of the Erg9 and Erg1 not result in a similar increase in azole resistance? Instead, we found that it results in a slightly increased azole susceptibility, which is in agreement with the previously reported synergistic interaction of terbinafine and azoles[26,27]. We believe that this is linked with the overall reduced synthesis of sterols which is expected when the enzymatic turnover by these enzymes is reduced. Because of this, the fungus will not only suffer from the accumulation of eburicol but also from a general depletion of sterols which in itself is incompatible with viability. In contrast, lowered expression of Erg6A would retain de novo synthesis of lanosterol, which may fulfill the functionality of the needed sterols to a certain degree, mitigating the deleterious effects of azole-induced eburicol accumulation[32].

Notably, in our study, we found that Erg6A is essential for the viability of *A. fumigatus*. In contrast, Erg6A homologues of yeasts are not essential for viability[22,33,34]. Consequently, the sterol C24-methyltransferase could represent a specific antifungal target in fungi that have an ergosterol biosynthesis pathway similar to that of *A. fumigatus*, similar as it was proposed recently[35].

A key question that arose in the present study was whether it is the accumulation of eburicol, or of the subsequently formed "toxic diol" which actually is responsible for the detrimental effects of azoles in *A. fumigatus*, including the formation of the cell wall patches. In baker's yeast and *Candida* species, the toxic effect of azoles is clearly attributed to the accumulation of the "toxic diol" as

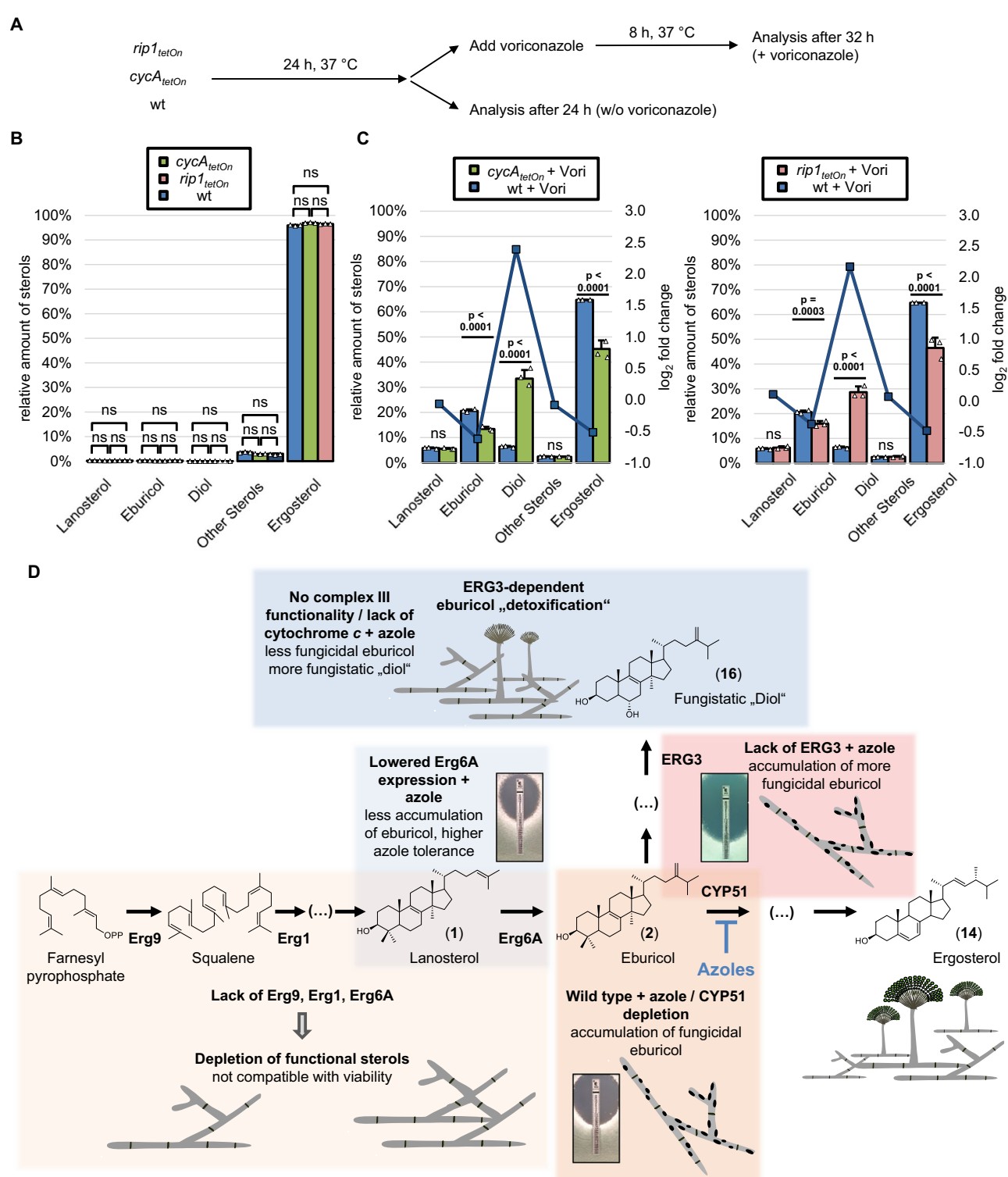

deletion of or mutations in *ERG3*, which is required to form it, results in azole resistance[14–17]. However, the conditional ERG3 triple mutant under repressed conditions still formed cell wall carbohydrate patches after azole exposure, even though it does not produce the "toxic diol". This demonstrates that neither ERG3 nor the "toxic diol" are required for the azole-induced cell wall patch formation. Furthermore, we show that a mutant lacking ERG3 is significantly more susceptible to azoles compared to the wild type. This indicates that the "toxic diol" which also accumulates in azole-treated *A. fumigatus* as a derivative of the toxic eburicol does not contribute to the toxic effect. In contrast, it even suggests that the conversion of eburicol to 14-methylergosta-8,24(28)-dien-3β,6α-diol ("toxic diol", **16**) contributes to the "detoxification" of eburicol. However, it cannot be excluded that the increased azole susceptibility of the ERG3 triple mutant under repressed conditions is linked to other effects, for example, to a more toxic effect of the eburicol when accumulating in combination with ergosta-7,22,24(28)-trien-3β-ol (**12**), 5-dihydroergosterol, or episterol (**11**). In either case, our results show that the toxicity of azoles in *Aspergillus* has a different mechanism than in baker's yeast and *Candida*.

**Fig. 7 | Increased formation of 14-methylergosta-8,24(28)-dien-3β,6α-diol in cytochrome *c*-depleted hyphae and model of the antifungal activity of azoles against *A. fumigatus*. A–C** Conidia of wild type (wt) and conditional mutants lacking cytochrome *c* (*cycA_{tetOn}*) or the Rieske protein (*rip1_{tetOn}*) in the absence of doxycycline were inoculated in Sabouraud liquid medium. After 24 h incubation in a rotary shake at 37 °C, mycelium was either directly harvested or supplemented with 1 μg ml⁻¹ voriconazole (+Vori), incubated in a rotary shake at 37 °C for another 8 h, and then harvested. For each condition, three biological replicates were cultured. **B, C** The sterol patterns of the harvested mycelia were analyzed by gas chromatography-mass spectrometry (GC-MS). The column graphs show the relative amounts (percentage of total sterol, left y-axis) of the indicated sterols for the indicated strains without voriconazole (**B**) or after exposure to voriconazole (**C**). The data points with the square symbols indicate the $\log_2$-fold change (right y-axis) in the amount of the respective sterol of the pairwise comparison of the conditions shown in the individual graphs. The $\log_2$-fold change data points were connected by lines for a better visual illustration of the changes in the profiles. Each column bar represents the mean of three replicates (data points) per condition, the error bars indicate standard deviations. The data represent one experiment that included two independent mutants with impaired mitochondrial complex III function (*cycA_{tetOn}*, *rip1_{tetOn}*). Statistical significance was set at $p < 0.05$, and calculated with a two-way ANOVA with Tukey's multiple comparison test. *p* values are indicated in the graphs; ns, not significant. Source data are provided as a Source Data file. **D** Model of the mode of action of azole antifungals in the pathogenic mold *A. fumigatus*. Azole antifungals inhibit sterol C14-demethylase (CYP51). This results in the accumulation of eburicol (2) which is fungicidal and triggers the formation of cell wall carbohydrate patches (area marked in orange). A lack of squalene synthase (Erg9), squalene epoxidase (Erg1) or sterol C24-methyltransferase (Erg6A) results in depletion of functional sterols, thus in suppression *A. fumigatus'* growth. However, this depletion does not result in the formation of the fungicidal cell wall carbohydrate patches (area marked in light orange). Lowered expression of Erg6A results in increased azole resistance because of reduced formation and, following azole treatment, reduced accumulation of eburicol (area marked in light blue). Following azole exposure, the accumulating eburicol is partially converted into the less toxic, fungistatic "diol" (14-methylergosta-8,24(28)-dien-3β,6α-diol; 16; area marked in blue). While in *A. fumigatus* wild type this conversion is less efficient, it is increased in *A. fumigatus* mutants that lack a functional mitochondrial complex III, explaining the fungistatic activity of azoles against and the reduced formation of cell wall carbohydrate patches in these mutants. A lack of sterol C5-desaturase (Erg3) does not result in increased resistance, but causes increased susceptibility to azole antifungals since the accumulating eburicol cannot be converted into the less toxic, fungistatic "diol" (area marked in red).

Interestingly, only low amounts of 14-methylfecosterol (**15**) accumulated in the azole-treated ERG3 mutant under repressed conditions. Similarly, unexpectedly low levels of 14-methylfecosterol (**15**) were also observed in *C. albicans* sterol C5-desaturase deletion mutants (Δ*erg3*/Δ*erg3*) after azole treatment or if CYP51 was deleted in parallel (Δ*erg11*/Δ*erg11*)[36,37]. A possible explanation for this could be that the accumulation of 14-methylergosta-8,24(28)-dien-3β-ol inhibits an upstream enzymatic step in the pathway.

Finally, we had previously shown that azole concentrations, which are normally fungicidal for *A. fumigatus*, become primarily fungistatic against *A. fumigatus* if the mitochondrial complex III is disrupted or not functional[7,31]. This correlates with a strikingly reduced formation of carbohydrate patches at these concentrations. The analysis of the sterol profiles of two mutants which lack mitochondrial complex III functionality, due to loss of the Rieske protein or cytochrome *c*, revealed that upon azole exposure the accumulating eburicol is much more efficiently converted to the "diol" (14-methylergosta-8,24(28)-dien-3β,6α-diol (**16**)) compared to azole-treated wild type. This supports our model, where the conversion of eburicol into 14-methylergosta-8,24(28)-dien-3β,6α-diol (**16**) helps *A. fumigatus* to survive lower azole concentrations and provides an explanation why *A. fumigatus* mutants with a dysfunctional mitochondrial complex III are more tolerant to azole antifungals. Furthermore, it supports the presumed fungistatic effects attributed to the accumulation of the "toxic diol", which is regularly observed in azole-treated baker's yeast and *Candida* species. In summary, our results demonstrate that the strong antifungal activity of azole antifungals on the fungal pathogen *A. fumigatus* relies on the specific accumulation of toxic eburicol which triggers the formation of cell wall carbohydrate patches that contribute to the strong fungicidal activity. Furthermore, our results suggest that alterations in the ergosterol biosynthesis pathway that lead to reduced eburicol accumulation relative to other sterols following azole treatment could contribute to azole resistance.

## Methods
### Strains, culture conditions, and chemicals
The nonhomologous end joining-deficient strain AfS35, a derivative of *A. fumigatus* D141[38,39], served as wild-type strain for all mutants used and constructed in this work. Conditional mutants and deletion mutants were constructed essentially as described before[20,39]. In all newly constructed deletion mutants, the respective genes were replaced by double-crossover homologous recombination using a self-excising hygromycin B resistance[40]. In all newly constructed conditional mutants, a doxycycline-inducible *pkiA-tetOn* promoter cassette[21]

was inserted upstream of the coding sequence of the respective genes. Mitochondria were visualized with a mitochondria-targeted green fluorescent protein (mtGFP) by transforming the respective strains with pCH005, essentially as described before[21]. All *A. fumigatus* strains used in this study are listed in Table 1. The *Candida* strains used were *C. albicans* ATCC14053 and *C. glabrata* ATCC2950. *Aspergillus* minimal medium (AMM)[41] and Sabouraud medium [4% (w/v) D-glucose, 1% (w/v) peptone (#LP0034; Thermo Fisher Scientific; Rockford, IL, US), pH 7.0 ± 0.2] were used in this study. Strains were generally maintained on AMM to obtain new conidia. Solid media were supplemented with 2% (w/v) agar (#214030; BD Bioscience, Heidelberg, Germany). All experiments involving growth tests were performed at least three times under similar conditions with concordant results if not stated differently. Calcofluor white (Fluorescent brightener 28; #ICNA0215806705) was obtained from VWR International (Radnor, PA, USA). Etest strips were obtained from bioMérieux (voriconazole; #412490; Marcy-l'Étoile, France). Doxycycline-hyclate was obtained from Sigma-Aldrich (#D9891-5G; St. Louis, MO, USA). Voriconazole was obtained from Biorbyt (#orb134756, Cambridge, East of England, United Kingdom), isavuconazole was obtained from Basilea (Basel, Switzerland), itraconazole (#I6657) and posaconazole (#32103) from Sigma-Aldrich. Terbinafin was obtained from Sigma-Aldrich (#T8826).

### Microscopy
Fluorescence microscopy was performed with a CSU-W1 Spinning Disk Field Scanning Confocal System (Yokogawa; Tokyo, Japan) mounted on an Eclipse Ti2 Inverted Microscope System (Nikon Instruments Inc; Melville, NY, USA) with the exception of the microscopy of terbinafine-treated hyphae, which was performed with a Lionheart FX automated microscope (Agilent BioTek; Santa Clara, CA, USA), and the microscopy of azole-treated yeasts and the respective *Aspergillus* control, which was performed with a Leica SP8 microscope (Leica Microsystems; Mannheim, Germany). The growth of mutants after exposure to different azoles was documented with a Lionheart FX automated microscope. For the confocal microscope, conidia were inoculated in 15 μ-Slide eight-well (#80826) slides (Ibidi; Martinsried, Germany) in 300 μl medium per well. Hyphae that were analyzed with the Lionheart FX automated microscope were inoculated in 96-well plates (#167008; Thermo Fisher Scientific; Rockford, IL, USA). Samples were generally incubated in humidified chambers to avoid evaporation of the medium. Cell wall chitin was stained with calcofluor white. When indicated, samples were fixed with 3.7% formaldehyde in ddH₂O for 3 min. These samples were then stained with 10 mg ml⁻¹ calcofluor white in ddH₂O for approximately 1 min, followed by washing the wells three times

**Table 1 | *A. fumigatus* strains relevant for or used in this work**

| Strain or genotype | Relevant genetic modification | Parental strain | Reference |
|---|---|---|---|
| AfS35 (wt) | *akuA::loxP* | D141 | 38,39 |
| *erg6A*$_{tetOn}$ | *(p)erg6A::ptrA-(p)pkiA-tetOn* | AfS35 (wt) | This work |
| *erg6A*$_{tetOn}$ + mtGFP | pCH005 | *erg6A*$_{tetOn}$ | This work |
| Δ*erg6B* | *erg6B::loxP-hygroR/blaster* | AfS35 (wt) | This work |
| *cyp51A*$_{tetOn}$ Δ*cyp51B* | *cyp51B::loxP-hygroR/blaster* | *cyp51A*$_{tetOn}$ | 7 |
| *cyp51A*$_{tetOn}$ | *(p)cyp51A::ptrA-(p)pkiA-tetOn* | AfS35 (wt) | 7 |
| *cyp51A*$_{tetOn}$ Δ*cyp51B* + mtGFP | pCH005 | *cyp51A*$_{tetOn}$ Δ*cyp51B* | 7 |
| *erg9*$_{tetOn}$ | *(p)erg9::ptrA-(p)pkiA-tetOn* | AfS35 (wt) | This work |
| *erg1*$_{tetOn}$ | *(p)erg1::ptrA-(p)pkiA-tetOn* | AfS35 (wt) | This work |
| wt + mtGFP | pCH005 | AfS35 (wt) | 7 |
| Δ*erg3A* | *erg3A::loxP-hygroR/blaster* | AfS35 (wt) | This work |
| Δ*erg3B* | *erg3B::loxP-hygroR/blaster* | AfS35 (wt) | This work |
| Δ*erg3C* | *erg3C::loxP-hygroR/blaster* | AfS35 (wt) | This work |
| *erg3A*$_{tetOn}$ | *(p)erg3A::ptrA-(p)pkiA-tetOn* | AfS35 (wt) | This work |
| *erg3A*$_{tetOn}$ Δ*erg3B* | *erg3B::loxP-hygroR/blaster* | *erg3A*$_{tetOn}$ | This work |
| *erg3A*$_{tetOn}$ Δ*erg3C* | *erg3C::loxP-hygroR/blaster* | *erg3A*$_{tetOn}$ | This work |
| *erg3C*$_{tetOn}$ | *(p)erg3C::ptrA-(p)pkiA-tetOn* | AfS35 (wt) | This work |
| *erg3C*$_{tetOn}$ Δ*erg3B* | *erg3B::loxP-hygroR/blaster* | *erg3C*$_{tetOn}$ | This work |
| *erg3A*$_{tetOn}$ Δ*erg3B* Δ*erg3C* | *erg3C::loxP-hygroR/blaster* | *erg3C*$_{tetOn}$ Δ*erg3B* | This work |
| *cycA*$_{tetOn}$ | *(p)cycA::ptrA-(p)pkiA-tetOn* | AfS35 (wt) | 7 |
| *rip1*$_{tetOn}$ | *(p)rip1::ptrA-(p)pkiA-tetOn* | AfS35 (wt) | 7 |

with ddH$_2$O. For live-cell microscopy and analysis of non-fixed samples, hyphae were stained by supplementing the medium with 3.33 µg ml$^{-1}$ calcofluor white for at least 5 min. The viability of hyphae expressing mitochondria-targeted GFP was analyzed by recording short time-lapse sequences with the spinning disc confocal microscopy, essentially as described before[7]. Hyphal compartments with evident mitochondrial dynamics were counted as viable. All microscopy-based experiments were performed at least three times under similar conditions with consistent results.

### Sterol analysis

Approximately $1 \times 10^6$ conidia were inoculated in 50 ml medium per flask and sample. For each condition, three biological replicates of the respective strain were cultivated and analyzed separately. Flasks were incubated in rotary shakers at 37 °C at 180 rounds per minute. To remove doxycycline, the mycelium of each sample was transferred into a Miracloth filter (#475855-1 R, Merck Millipore, Burlington, MA, USA) fitted in a funnel and washed three times with medium. The mycelium was then transferred into a new bottle with fresh medium. To harvest mycelium for analysis by gas chromatography-mass spectrometry (GC-MS), mycelium was transferred into a Miracloth filter in a funnel, washed three times with phosphate-buffered saline, frozen in liquid nitrogen, crushed with a mortar and pestle and stored at −80 °C. The thawed wet fungal biomass was subsequently used for sterol extraction as described by Müller et al.[11,42].

The sterols were analyzed as their corresponding trimethylsilyl (TMS) ethers by GC-MS, essentially as described before[11]. Overall, 17 different sterols were detected, of which 16 were identified. The sterol TMS ethers were identified by mass spectra and relative retention times (RRT). The base peak of each sterol TMS ether were taken as a quantifier ions for calculating the peak areas for internal standard (IS) cholestane *m/z* 217, RRT 1.00; ergosta-8,22,24(28)-trien-3β-ol *m/z* 253, RRT 1.28; ergosta-5,8,22-trien-3β-ol (lichesterol) *m/z* 363, RRT 1.30; ergosta-5,8,22,24(28)-tetraen-3β-ol *m/z* 361, RRT 1.31; ergosta-5,7,22-trien-3β-ol (ergosterol, **14**) *m/z* 363 RRT 1.33; ergosta-7,22-dien-3β-ol (5-dihydroergosterol) *m/z* 343, RRT 1.36; ergosta-5,7,22,24(28)-tetraen-3β-ol (dehydroergosterol, **13**) *m/z* 361, RRT 1.36; ergosta-7,22,24(28)-

trien-3β-ol (**12**) *m/z* 343, RRT 1.38; ergosta-5,7-dien-3β-ol *m/z* 365, RRT 1.40; ergosta-7,24(28)-dien-3β-ol (episterol, **11**) *m/z* 343, RRT 1.41; 14-methylergosta-8,24(28)-dien-3β,6α-diol ("toxic diol", **16**) *m/z* 377, RRT 1.42; 4,4,14-trimethylcholesta-8,24(25)-dien-3β-ol (lanosterol, **1**) *m/z* 393, RRT 1.44; 4-methylergosta-8,24(28)-dien-3β-ol (4-methylfecosterol, **5**) *m/z* 379, RRT 1.46; 4,4,14-trimethylergosta-8,24(28)-dien-3β-ol (eburicol, **2**) *m/z* 407, RRT 1.50; 4,4-dimethylergosta-8,24(28)-dien-3β-ol (**4**) *m/z* 408, RRT 1.53; 14-methylergosta-8,24(28)-dien-3β-ol (14-methylfecosterol, **15**) *m/z* 379, RRT 1.35; ergosta-X,Y-dien-3β-ol (unidentified sterol) *m/z* 255, RRT 1.30; cholesterol (detected in traces (<0.5% of the total sterols); considered a contamination from the growth medium, not shown in the tables) *m/z* 368.5, RRT 1.24. The peak area of each sterol TMS ether was determined, and the percentage of sterol was calculated[10,43]. Statistical significance was calculated with GraphPad Prism (Version 10.1.1 (Prism 10); Dotomatics, Boston, MA, USA) was calculated with a two-way ANOVA with Tukey's multiple comparison test for all sterols.

### Use of generative artificial intelligence

DeepL (deepl.com; Cologne, Germany) was used to identify and correct errors in spelling, grammar, and punctuation in the manuscript texts.

### Reporting summary

Further information on research design is available in the Nature Portfolio Reporting Summary linked to this article.

## Data availability

The data that support the findings of this study are included in this article and its Supplementary Information files, in the Source Data file or on request from the corresponding author. Source data are provided with this paper.

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

## Acknowledgements

This work was in part supported by the German Research Foundation (430055013; J.W.), the Irish Health Research Board (SS-2022-016; N.N.), the Förderprogramm für Forschung und Lehre (FöFoLe) of the Medical Faculty of the LMU München (49/2016; I.K. and J.W.), and the Graduate School of Life Sciences (GSLS) at the University of Würzburg (E.G.). C.M. thanks Franz Bracher for providing his laboratories and equipment.

## Author contributions

J.W. conceived the study. J.W., H.E., E.G., S.B., and C.M. planned the experiments. H.E., E.G., I.K., K.D., and J.W. constructed the mutant strains. H.E., E.G., S.B., N.N., and R.M. performed experiments. C.M. performed the sterol analyses. J.W., H.E., E.G., and C.M. wrote the manuscript draft. All authors reviewed and authorized the manuscript.

## Funding

## Competing interests

J.W. declares that he has received research grants and financial support for the organization of educational events from Pfizer and speaker honoraria for lectures from Pfizer and Gilead. The remaining authors declare no competing interests.
