## [Peer Review File · Nature Communications]

Toxic eburicol accumulation drives the antifungal activity of azoles against *Aspergillus fumigatus*REVIEWER COMMENTS

Reviewer #1 (Remarks to the Author):

The article entitled "Toxic eburicol accumulation drives the antifungal activity of azoles against *Aspergillus fumigatus*", is an interesting study that elucidates a new model of for azole Drugs against *Aspergillus fumigatus*.

The topic discussed in this manuscript holds significant potential interest, particularly given the current limitations in the number of antifungals available to effectively treat *Aspergillus* infections. Moreover, the emergence of antifungal drug resistance presents a persistent challenge in clinical practice, underscoring the urgent need for evolving treatment options. Understanding the mechanisms of action of azoles, as well as the biosynthesis pathway of ergosterol, is therefore crucial for both investigating azole-resistant mechanisms and identifying novel targets for antifungal therapies.

Overall, the manuscript is very clear, although some points should be addressed in order to accept the article for publication:

- According to this Study the mode of action of azoles against *A. fumigatus* relies on the accumulation of eburicol and the formation of cell Wall carbohydrate patches. Do the authors have made any hypothesis about the mechanism in resistant strains?
- Previous studies have identified C-24 ethyl sterols in *A. fumigatus*. C-24 alkylation is catalysed by S-adenosyl-methionine-sterol-C-methyltransferases (SAMs) and at least two methyltransferases are involved in consecutive methylation reactions leading to 24-ethyl sterols in higher plants. In plants C-24 ethyl sterols have multiple roles to play in growth and development, however, few reports exist on the detection of 24-ethyl sterols in fungi, and their role in *A. fumigatus* remains unknown. Have the authors investigated any role for Erg6A and/or Erg6B on that?. Did you detect ethyl sterols in the mutant strains?
- Lines 184-186....I found this statement very interesting but a little vague and certainly needs further explanations.

Other points are:

Line 79 please refer to *Saccharomyces cerevisiae* as baker's yeast the first time that Baker's yeast is used in the text.

Line 85 the number assigned for ergosterol (12) is wrong and should be 14.

Line 395 correct 10 mg ml⁻¹

Figure 2E: MICs values are not correct for some of the strains.

Figure 3D the same mistake tan for Figure 2E.

Figure 6B MIC values are not shown in the figure.

Reviewer #2 (Remarks to the Author):

General comment: In this study, Elsaman et al. decipher the ergosterol biosynthesis pathway (target of azole drugs) in *Aspergillus fumigatus*, and show distinct features compared to that of yeasts, which may explain the distinct effects of azole drugs against yeasts and molds. This is a very important study, as it addresses a hot topic of mycology (azole resistance) and may suggest new antifungal targets for novel mold-specific antifungals. The methodological approach is robust and very complete. The manuscript is well presented. I have only minor comments.

Paragraph lines 164-179 and Figure 1: I do not see ERG1 and ERG9 in the Figure. Therefore, it is not easy to understand where they are in the pathway.

Figure 1 is important, but somewhat busy and complex. Not sure it is important to show the molecular structure of the compounds. It could be completed (or shown in a distinct figure) how this impacts on azole susceptibility, toxic sterol accumulation and cell wall patch formation in a schematic representation.

Figures 2E and 3E: I would suggest to show this with a spotting assay.

Figure 4, 5, 7 and Suppl 1: You show the proportions of the different sterols. Maybe there is also an interest to show differences in the total biomass for ergosterol and some other sterols (?)

Although this is expected that you will see the same effects, did you test the susceptibility of your mutants to mold-active azoles other than voriconazole ?

Reviewer #3 (Remarks to the Author):

In the manuscript NCOMMS-24-09584 "Toxic eburicol accumulation drives the antifungal activity of azoles against *Aspergillus fumigatus*" The authors show that the accumulation of carbohydrate patches after azole treatment does not occur without a functioning Erg6A. They therefore propose that the fungicidal effect of azoles in *A. fumigatus* is due to the accumulation of eburicol (and resultant formation of carbohydrate patches in the cell wall) and not the accumulation of the 'toxic sterol' (14 methyl ergosta-8,24(28)-dien-3,6-diol) as seen in other fungal species. This work uses a range of strains with regulatable genes to neatly isolate the effect of azoles and alterations in sterol biosynthesis on the growth of (and accumulation of carbohydrate patches in) *A. fumigatus*. Of importance, it shows that carbohydrate patches occur observed when there is an accumulation of eburicol, rather than when the 'toxic sterol' and that this effect is not due to a reduction in sterols overall (as compared with ERG1 and ERG9 mutant strains), The manuscript is well written and generally clear and the work clearly shows there is a difference in underlying mode of action of azoles in *A. fumigatus* compared with *Candida* and other yeasts, This is of importance in understanding the action of and resistance to antifungals.

Comments;

- Line 243, the 'other sterols' accumulated in the azole-exposed ERG3 mutant are all 14 demethylated. Why was 14 methyl ergosta-8,24(28)-dienol not seen in these strains when treated with azole? If Erg3 activity is responsible for the production of the 'toxic sterol' it would be expected that this sterol accumulates alongside the lanosterol and eburicol.
- Throughout the manuscript the azole treatment of yeasts is said to result in the accumulation of the 'toxic sterol' (14 methyl ergosta-8,24(28)-dien-3,6-diol) when yeasts are treated with azoles. However, this is not always the case, *Cryptococcus* spp. accumulate sterones (eburicone and lanosterolne) upon azole treatment.
- The rationale of investigating mutants lacking mitochondrial complex III functionality was not entirely clear. Why were these particular genes chosen?
- Line 213 - The presence of toxic diol after azole treatment has previously been reported in *A. fumigatus* and therefore this finding should also have been expected
- Figure 1 A and B – appear to be missing the 2 other genes in the C4 demethylase complex; Afuerg26 and Afuerg27.

Response to the reviewers' comments

We would like to thank the reviewers for their very positive feedback. We have revised the manuscript according to the very helpful reviewers' comments and suggestions and hope that it is now acceptable for publication in *Nature Communications*.

Please find our point-by-point response to the reviewers' comments below.

Reviewer #1

The article entitled "Toxic eburicol accumulation drives the antifungal activity of azoles against *Aspergillus fumigatus*", is an interesting study that elucidates a new model of for azol Drugs against *Aspergillus fumigatus*.

The topic discussed in this manuscript holds significant potential interest, particularly given the current limitations in the number of antifungals available to effectively treat *Aspergillus* infections. Moreover, the emergence of antifungal drug resistance presents a persistent challenge in clinical practice, underscoring the urgent need for evolving treatment options. Understanding the mechanisms of action of azoles, as well as the biosynthesis pathway of ergosterol, is therefore crucial for both investigating azole-resistant mechanisms and identifying novel targets for antifungal therapies.

Overall, the manuscript is very clear, although some points should be addressed in order to accept the article for publication:

We are very grateful for the very positive evaluation of our manuscript by this Reviewer. Please see our responses to the individual points below.

- According to this Study the mode of action of azols againts *A. fumigatus* relies on the accumulation of eburicol and the formation of cell Wall carbohydrate patches. Do the authors have made any hypothesis about the mechanism in resistant strains?

This is an interesting point. The most common and best characterized azole resistance mechanisms in *A. fumigatus* are mutations that result in 1) overexpression or 2) alteration of the target enzyme. In other cases, 3) overexpression of efflux pumps or 4) mutations in various other genes (for example, *hapE*, *hmg1*, *srbA*) which presumably affect expression or activity of ergosterol biosynthesis enzymes have been linked with azole resistance (*J Antimicrob Chemother.* 2022;77(8):2053-2073, <https://doi.org/10.1093/jac/dkac161>).

The resistance mechanisms 1), 2), and 3) are all clearly based on the upkeep of the sterol C14-demethylase activity of CYP51. This would prevent eburicol from accumulating and thus, according to our model, prevent formation of cell wall carbohydrate patches. In agreement with this, azole-resistant *A. fumigatus* isolates which were obtained from a clinical strain collection

did not form cell wall patches when we treated them with azoles (Nat Commun. 2018;9(1):3098, <https://10.1038/s41467-018-05497-7>).

It is more difficult to discuss the remaining mechanisms that belong to 4), as these have not yet been investigated in detail. Indeed, in these cases the mechanisms could be based on mutations that trigger alterations in the ergosterol biosynthesis pathway causing reduced eburicol accumulation relative to other sterols following azole treatment, either by reducing its de novo synthesis, similar to what we saw upon repression of *erg6A*, or by enhancing its conversion to less toxic sterols such as 14-methylergosta-8,24(28)-dien-3 β ,6 α -diol. To address this important point, we now mention this in the conclusion of our manuscript (lines 377-380).

- Previous studies have identified C-24 ethyl sterols in *A. fumigatus*. C-24 alkylation is catalysed by S-adenosyl-methionine-sterol-C-methyltransferases (SAMs) and at least two methyltransferases are involved in consecutive methylation reactions leading to 24-ethyl sterols in higher plants. In plants C-24 ethyl sterols have multiple roles to play in growth and development, however, few reports exist on the detection of 24-ethyl sterols in fungi, and their role in *A. fumigatus* remains unknown.

Have the authors investigated any role for Erg6A and/or Erg6B on that?. Did you detect ethyl sterols in the mutant strains?

Thank you for this question. We have carefully reviewed the sterol profiles of the experiments again and did not detect ethyl sterols in any of the mutants. The analytical method used is highly sensitive and can detect phytosterols, as demonstrated in previous work of Christoph Müller (Planta Med. 2015;81(7):613-20, <https://doi.org/10.1055/s-0035-1545906>). If ethyl sterols were formed in significant amounts in our mutants, we would have detected them. Our results therefore indicate that they are not formed in significant amounts in our mutants under the selected conditions. To address this point, we now provide tables that show all sterols detected in the analyses (see new Supplementary Tables 1-4).

- Lines 184-186...I found this statement very interesting but a little vague and certainly needs further explanations.

Thank you very much for pointing this out, we agree that this statement was vague and have carefully revised the paragraph to better explain our findings (lines 180-187 in the revised manuscript). To support our findings, we have added new Supplementary Figure 3 and mention the finding of two other studies in which it was shown that the Erg1 inhibitor terbinafine and azoles act synergistically on *A. fumigatus* (Med Mycol. 2001;39(1):91-5, <https://doi.org/10.1080/mmy.39.1.91.95>; J Antimicrob Chemother. 2002;50(2):189-94, <https://doi.org/10.1093/jac/dkf111>).

Other points are:

Line 79 please refer to *Saccharomyces cerevisiae* as baker's yeast the first time that Baker's yeast is used in the text.

Thank you, we have corrected this.

Line 85 the number assigned for ergosterol (12) is wrong and should be 14.

Thank you for noticing this, we have changed it accordingly.

Line 395 correct 10 mg ml "-1"

Corrected.

Figure 2E: MICs values are not correct for some of the strains.

The concentrations indicated in the panel show the amount of doxycycline that was used to induce the Tet-On promoter. To indicate this more clearly, we have now supplemented a corresponding label (Doxy) in the panels of the revised figures.

Figure 3D the same mistake tan for Figure 2E.

Figure 6B MIC values are not shown in the figure.

Thank you, as stated above: we have now supplemented a corresponding label for this in the panels of the revised figures.

Reviewer #2

General comment: In this study, Elsaman et al. decipher the ergosterol biosynthesis pathway (target of azole drugs) in *Aspergillus fumigatus*, and show distinct features compared to that of yeasts, which may explain the distinct effects of azole drugs against yeasts and molds. This is a very important study, as it addresses a hot topic of mycology (azole resistance) and may suggest new antifungal targets for novel mold-specific antifungals. The methodological approach is robust and very complete. The manuscript is well presented. I have only minor comments.

We would also like to thank this reviewer for his very positive evaluation of our manuscript.

Paragraph lines 164-179 and Figure 1: I do not see ERG1 and ERG9 in the Figure. Therefore, it is not easy to understand where they are in the pathway.

We fully agree and have revised Figure 1, which now illustrates the positions and roles of Erg1 and Erg9 in the pathway.

Figure 1 is important, but somewhat busy and complex. Not sure it is important to show the molecular structure of the compounds. It could be completed (or shown in a distinct figure) how this impacts on azole susceptibility, toxic sterol accumulation and cell wall patch formation in a schematic representation.

We fully agree with the reviewer that the illustration of the sterol biosynthesis pathway is too busy. To address this point, we have revised Figure 1, in which we have removed the molecular structures of the compounds, better illustrate how the azole alters the pathways, and have labelled the 14-methylergosta-8,24(28)-dien-3 β ,6 α -diol (16) as “toxic diol”. The illustrations with the molecular structures, which now also shows the positions and roles of Erg1 and Erg9, is now provided as new Supplementary Figure 1.

How azoles impact on the cell wall patch formation and general toxicity is shown separately in Figure 7 D.

Figures 2E and 3E: I would suggest to show this with a spotting assay.

Following the reviewer’s suggestion, we have performed spotting assays with the *erg6A* mutant, the ERG3 triple mutant, and the *erg1* mutant. The data are now presented together with the growth tests performed with different mold-active azoles in Supplementary Figures 2, 3 and 5.

Figure 4, 5, 7 and Suppl 1: You show the proportions of the different sterols. Maybe there is also an interest to show differences in the total biomass for ergosterol and some other sterols (?)

When we carried out the experiments, we were in the first place interested in the relative amounts of sterols. Therefore, a lyophilization step was omitted to reduce the logistic complexity of the experiments. Instead, wet mycelium was used for the analysis of the sterols, which did not allow for accurate determination of the mycelium biomasses. For this reason, we unfortunately do not have exact mycelium masses with which we could calculate these values.

Although this is expected that you will see the same effects, did you test the susceptibility of your mutants to mold-active azoles other than voriconazole ?

To answer the reviewer's question, we have performed growth tests with other mold-active azoles. These results show that the observed changes in the susceptibilities of the mutants also apply to other azoles such as itraconazole, isavuconazole and posaconazole (see Supplementary Figures 2, 3 and 5).

Reviewer #3

In the manuscript NCOMMS-24-09584 "Toxic eburicol accumulation drives the antifungal activity of azoles against *Aspergillus fumigatus*" The authors show that the accumulation of carbohydrate patches after azole treatment does not occur without a functioning Erg6A. They therefore propose that the fungicidal effect of azoles in *A. fumigatus* is due to the accumulation of eburicol (and resultant formation of carbohydrate patches in the cell wall) and not the accumulation of the 'toxic sterol' (14 methyl ergosta-8,24(28)-dien-3,6-diol) as seen in other fungal species.

This work uses a range of strains with regulatable genes to neatly isolate the effect of azoles and alterations in sterol biosynthesis on the growth of (and accumulation of carbohydrate patches in) *A. fumigatus*. Of importance, it shows that carbohydrate patches occur observed when there is an accumulation of eburicol, rather than when the 'toxic sterol' and that this effect is not due to a reduction in sterols overall (as compared with ERG1 and ERG9 mutant strains),

The manuscript is well written and generally clear and the work clearly shows there is a difference in underlying mode of action of azoles in *A. fumigatus* compared with *Candida* and other yeasts, This is of importance in understanding the action of and resistance to antifungals.

We are also very grateful for the very positive evaluation of our manuscript by Reviewer #3.

Comments;

- Line 243, the 'other sterols' accumulated in the azole-exposed ERG3 mutant are all 14 demethylated. Why was 14 methyl ergosta-8,24(28)-dienol not seen in these strains when treated with azole? If Erg3 activity is responsible for the production of the 'toxic sterol' it would be expected that this sterol accumulates alongside the lanosterol and eburicol.

Thank you for this important comment. We looked into this. Indeed, in the experiment depicted in Figure 5 C and D (see new Supplementary Table 3), 14-methylergosta-8,24(28)-dien-3 β -ol (14-methylfecosterol) gets formed both in wild type and the ERG3 mutant under repressed conditions after azole treatment, but only in very low amounts (approx. 0.3%). Similarly low levels (<1%) were detected in two other independent experiments performed under similar conditions with a second *A. fumigatus* ERG3 mutant.

While we do not have a fully satisfying explanation for the relatively low accumulation, a similar phenomenon has also been noted in *Candida albicans*: Interestingly, surprisingly small amounts of 14-methylergosta-8,24(28)-dien-3 β -ol (14-methylfecosterol) also accumulate in *C. albicans* *erg3 Δ / Δ* mutants (approx. 4%) when treated with an azole (**Antimicrob Agents Chemother.** 2021;65(12):e0104421, <https://doi.org/10.1128/aac.01044-21>) or when CYP51 is additionally deleted (**Antimicrob Agents Chemother.** 2003 Aug;47(8):2404-12, <https://doi.org/10.1128/aac.47.8.2404->

2412.2003). Again, the corresponding Erg3-expressing controls tend to accumulate similar amounts or even more 14-methylfecosterol, very similar to what we observed in *A. fumigatus*. In our opinion, the most plausible hypothesis explaining these findings is that 14-methylergosta-8,24(28)-dien-3 β -ol (14-methylfecosterol) inhibits an enzyme upstream in the pathway (e.g., *AfErg25A/B*, *AfErg26A/B* or *AfErg27*). An alternative explanation could be that in *A. fumigatus* eburicol is first processed by ERG3 and in the second step by ERG25/26/27.

To address this point, we now mention the low amounts of 14-methylergosta-8,24(28)-dien-3 β -ol (14-methylfecosterol) in the manuscript (lines 252-255 and lines 356-361) and provide supplementary tables with the quantities of all detected sterols for the experiments (see new Supplementary Tables 1 – 4).

- Throughout the manuscript the azole treatment of yeasts is said to result in the accumulation of the 'toxic sterol' (14 methyl ergosta-8,24(28)-dien-3,6-diol) when yeasts are treated with azoles. However, this is not always the case, *Cryptococcus* spp. accumulate sterones (eburicone and lanosterol) upon azole treatment.

Thank you for highlighting this important fact. We have revised the manuscript throughout and replaced such statements to avoid ambiguity.

- The rationale of investigating mutants lacking mitochondrial complex III functionality was not entirely clear. Why were these particular genes chosen?

In our previous study (*Nat Commun.* 2018;9(1):3098, <https://10.1038/s41467-018-05497-7>) we reported the unexpected finding that mutants lacking mitochondrial complex III functionality are susceptible to azoles, but the azoles partially lose their fungicidal activity, which correlates with reduced formation of cell wall carbohydrate patches. We could not fully explain the reduced fungicidal activity in our previous work. Our novel insights could provide an explanation of this phenotype. Indeed, we show that less eburicol accumulates in these mutants after azole treatment. To address this point, we have added a sentence to the respective paragraph to explain that the reason for the reduced fungicidal activity remained unexplained in our previous work and the experiment was conducted to test if the new findings could provide an explanation for the increased survival of the mitochondrial complex III functionality mutants (lines 283-284).

- Line 213 - The presence of toxic diol after azole treatment has previously been reported in *A. fumigatus* and therefore this finding should also have been expected

Thank you for bringing this to our attention. We found a study that had reported formation of toxic diol previously (*Int J Antimicrob Agents.* 2019;54(4):449-455,

<https://doi.org/10.1016/j.ijantimicag.2019.07.011>) and now mention this study in the revised manuscript (lines 214-215).

- Figure 1 A and B – appear to be missing the 2 other genes in the C4 demethylase complex; Afuerg26 and Afuerg27.

Thank you for spotting this, we have added the enzymes to the revised figures (Figure 1 and Supplementary Fig. 1).